# Prediction Poisoning: Towards Defenses Against DNN Model Stealing Attacks

**Tribhuvanesh Orekondy[1], Bernt Schiele[1], Mario Fritz[2]**
[1] Max Planck Institute for Informatics
[2] CISPA Helmholtz Center for Information Security
Saarland Informatics Campus, Germany
{orekondy, schiele}@mpi-inf.mpg.de, fritz@cispa.saarland

## Abstract

High-performance Deep Neural Networks (DNNs) are increasingly deployed in many real-world applications e.g., cloud prediction APIs. Recent advances in model functionality stealing attacks via black-box access (i.e., inputs in, predictions out) threaten the business model of such applications, which require a lot of time, money, and effort to develop. Existing defenses take a passive role against stealing attacks, such as by truncating predicted information. We find such passive defenses ineffective against DNN stealing attacks. In this paper, we propose the first defense which actively perturbs predictions targeted at poisoning the training objective of the attacker. We find our defense effective across a wide range of challenging datasets and DNN model stealing attacks, and additionally outperforms existing defenses. Our defense is the first that can withstand highly accurate model stealing attacks for tens of thousands of queries, amplifying the attacker's error rate up to a factor of $85\times$ with minimal impact on the utility for benign users.

## 1 Introduction

Effectiveness of state-of-the-art DNN models at a variety of predictive tasks has encouraged their usage in a variety of real-world applications e.g., home assistants, autonomous vehicles, commercial cloud APIs. Models in such applications are valuable intellectual property of their creators, as developing them for commercial use is a product of intense labour and monetary effort. Hence, it is vital to preemptively identify and control threats from an adversarial lens focused at such models. In this work we address model stealing, which involves an adversary attempting to counterfeit the functionality of a target victim ML model by exploiting black-box access (query inputs in, posterior predictions out).

Stealing attacks dates back to Lowd & Meek (2005), who addressed reverse-engineering linear spam classification models. Recent literature predominantly focus on DNNs (specifically CNN image classifiers), and are shown to be highly effective (Tramèr et al., 2016) on complex models (Orekondy et al., 2019), even without knowledge of the victim's architecture (Papernot et al., 2017b) nor the training data distribution. The attacks have also been shown to be highly effective at replicating pay-per-query image prediction APIs, for as little as $30 (Orekondy et al., 2019).

Defending against stealing attacks however has received little attention and is lacking. Existing defense strategies aim to either *detect* stealing query patterns (Juuti et al., 2019), or degrade quality of predicted posterior via *perturbation*. Since detection makes strong assumptions on the attacker's query distribution (e.g., small $L_2$ distances between successive queries), our focus is on the more popular perturbation-based defenses. A common theme among such defenses is accuracy-preserving posterior perturbation: the posterior distribution is manipulated while retaining the top-1 label. For instance, rounding decimals (Tramèr et al., 2016), revealing only high-confidence predictions (Orekondy et al., 2019), and introducing ambiguity at the tail end of the posterior distribution (Lee et al., 2018). Such strategies benefit from preserving the accuracy metric of the defender. However, in line with previous works (Tramèr et al., 2016; Orekondy et al., 2019; Lee et al., 2018), we find models can be effectively stolen using just the top-1 predicted label returned by the black-box. Specifically, in many cases we observe <1% difference between attacks that use the full range of

posteriors (blue line in Fig. 1) to train stolen models and the top-1 label (orange line) alone. In this paper, we work towards effective defenses (red line in Fig. 1) against DNN stealing attacks with minimal impact to defender's accuracy.

The main insight to our approach is that unlike a benign user, a model stealing attacker additionally uses the predictions to *train* a replica model. By introducing controlled perturbations to predictions, our approach targets poisoning the training objective (see Fig. 2). Our approach allows for a utility-preserving defense, as well as trading-off a marginal utility cost to significantly degrade attacker's performance. As a practical benefit, the defense involves a single hyperparameter (perturbation utility budget) and can be used with minimal overhead to any classification model without retraining or modifications.

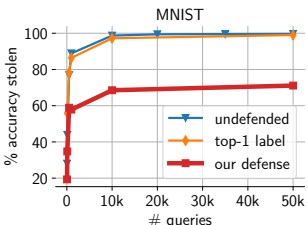

**Figure 1:** We find existing defenses (orange line) ineffective against recent attacks. Our defense (red line) in contrast significantly mitigates the attacks.

We rigorously evaluate our approach by defending six victim models, against four recent and effective DNN stealing attack strategies (Papernot et al., 2017b; Juuti et al., 2019; Orekondy et al., 2019). Our defense consistently mitigates all stealing attacks and further shows improvements over multiple baselines. In particular, we find our defenses degrades the attacker's query sample efficiency by 1-2 orders of magnitude. Our approach significantly reduces the attacker's performance (e.g., 30-53% reduction on MNIST and 13-28% on CUB200) at a marginal cost (1-2%) to defender's test accuracy. Furthermore, our approach can achieve the same level of mitigation as baseline defenses, but by introducing significantly lesser perturbation.

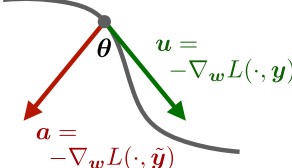

Our Perturbation Objective:

$$\underset{\tilde{y}}{\arg\max}\ \boldsymbol{\theta} \quad \text{s.t} \quad \text{dist}(\boldsymbol{y}, \tilde{\boldsymbol{y}}) \leq \epsilon$$

**Figure 2:** We perturb posterior predictions $\tilde{\boldsymbol{y}} = \boldsymbol{y} + \boldsymbol{\delta}$, with an objective of poisoning the adversary's gradient signal.

**Contributions.** (i) We propose the first utility-constrained defense against DNN model stealing attacks; (ii) We present the first active defense which poisons the attacker's training objective by introducing bounded perturbations; and (iii) Through extensive experiments, we find our approach consistently mitigate various attacks and additionally outperform baselines.

## 2 RELATED LITERATURE

*Model stealing attacks* (also referred to as 'extraction' or 'reverse-engineering') in literature aim to infer hyperparameters (Oh et al., 2018; Wang & Gong, 2018), recover exact parameters (Lowd & Meek, 2005; Tramèr et al., 2016; Milli et al., 2018), or extract the functionality (Correia-Silva et al., 2018; Orekondy et al., 2019) of a target black-box ML model. In some cases, the extracted model information is optionally used to perform evasion attacks (Lowd & Meek, 2005; Nelson et al., 2010; Papernot et al., 2017b). The focus of our work is model *functionality* stealing, where the attacker's yardstick is test-set accuracy of the stolen model. Initial works on stealing simple linear models (Lowd & Meek, 2005) have been recently succeeded by attacks shown to be effective on complex CNNs (Papernot et al., 2017b; Correia-Silva et al., 2018; Orekondy et al., 2019) (see Appendix B for an exhaustive list). In this work, we works towards defenses targeting the latter line of DNN model stealing attacks.

Since ML models are often deployed in untrusted environments, a long line of work exists on guaranteeing certain (often orthogonal) properties to safeguard against malicious users. The properties include *security* (e.g., robustness towards adversarial evasion attacks (Biggio et al., 2013; Goodfellow et al., 2014; Madry et al., 2018)) and *integrity* (e.g., running in untrusted environments (Tramer & Boneh, 2019)). To prevent leakage of *private* attributes (e.g., identities) specific to training data in the resulting ML model, *differential privacy* (DP) methods (Dwork et al., 2014) introduce randomization during training (Abadi et al., 2016; Papernot et al., 2017a). In contrast, our defense objective is to provide *confidentiality* and protect the functionality (intellectual property) of the ML model against illicit duplication.

*Model stealing defenses* are limited. Existing works (which is primarily in multiclass classification settings) aim to either *detect* stealing attacks (Juuti et al., 2019; Kesarwani et al., 2018; Nelson et al., 2009; Zheng et al., 2019) or *perturb* the posterior prediction. We focus on the latter since detection involves making strong assumptions on adversarial query patterns. Perturbation-based defenses are predominantly non-randomized and accuracy-preserving (i.e., top-1 label is unchanged). Approaches include revealing probabilities only of confident classes (Orekondy et al., 2019), rounding probabilities (Tramèr et al., 2016), or introducing ambiguity in posteriors (Lee et al., 2018). None of the existing defenses claim to mitigate model stealing, but rather they only marginally delay the attack by increasing the number of queries. Our work focuses on presenting an effective defense, significantly decreasing the attacker's query sample efficiency within a principled utility-constrained framework.

## 3 PRELIMINARIES

**Model Functionality Stealing.**    Model stealing attacks are cast as an interaction between two parties: a victim/defender $V$ ('teacher' model) and an attacker $A$ ('student' model). The only means of communication between the parties are via black-box queries: attacker queries inputs $\boldsymbol{x} \in \mathcal{X}$ and defender returns a posterior probability distribution $\boldsymbol{y} \in \Delta^K = P(\boldsymbol{y}|\boldsymbol{x}) = F_V(\boldsymbol{x})$, where $\Delta^K = \{\boldsymbol{y} \succeq 0, \mathbf{1}^T \boldsymbol{y} = 1\}$ is the probability simplex over $K$ classes (we use $K$ instead of $K-1$ for notational convenience). The attack occurs in two (sometimes overlapping) phases: (i) *querying*: the attacker uses the black-box as an oracle labeler on a set of inputs to construct a 'transfer set' of input-prediction pairs $\mathcal{D}^{\text{transfer}} = \{(\boldsymbol{x}_i, \boldsymbol{y}_i)\}_{i=1}^B$; and (ii) *training*: the attacker trains a model $F_A$ to minimize the empirical risk on $\mathcal{D}^{\text{transfer}}$. The end-goal of the attacker is to maximize accuracy on a held-out test-set (considered the same as that of the victim for evaluation purposes).

**Knowledge-limited Attacker.**  In model stealing, attackers justifiably lack complete knowledge of the victim model $F_V$. Of specific interest are the model architecture and the input data distribution to train the victim model $P_V(X)$ that are not known to the attacker. Since prior work (Hinton et al., 2015; Papernot et al., 2016; Orekondy et al., 2019) indicates functionality largely transfers across architecture choices, we now focus on the query data used by the attacker. Existing attacks can be broadly categorized based on inputs $\{\boldsymbol{x} \sim P_A(X)\}$ used to query the black-box: (a) *independent distribution*: (Tramèr et al., 2016; Correia-Silva et al., 2018; Orekondy et al., 2019) samples inputs from some distribution (e.g., ImageNet for images, uniform noise) independent to input data used to train the victim model; and (b) *synthetic set*: (Papernot et al., 2017b; Juuti et al., 2019) augment a limited set of seed data by adaptively querying perturbations (e.g., using FGSM) of existing inputs. We address both attack categories in our paper.

**Defense Objectives.**    We perturb predictions in a controlled setting: $\tilde{\boldsymbol{y}} = F_V^{\delta}(\boldsymbol{x}) = \boldsymbol{y} + \boldsymbol{\delta}$ s.t. $\tilde{\boldsymbol{y}}, \boldsymbol{y} \in \Delta^K$. The defender has two (seemingly conflicting) objectives: (i) **utility**: such that perturbed predictions remain useful to a benign user. We consider two utility measures: (a) $\text{Acc}(F_V^{\delta}, \mathcal{D}^{\text{test}})$: accuracy of defended model on test examples; and (b) $\text{dist}(\boldsymbol{y}, \tilde{\boldsymbol{y}}) = ||\boldsymbol{y} - \tilde{\boldsymbol{y}}||_p = \epsilon$ to measure perturbation. (ii) **non-replicability**: to reduce the test accuracy of an attacker (denoted as $\text{Acc}(F_A, \mathcal{D}^{\text{test}})$) who exploits the predictions to train a replica $F_A$ on $\mathcal{D}^{\text{transfer}}$. For consistency, we evaluate both the defender's and attacker's stolen model accuracies on the same set of test examples $\mathcal{D}^{\text{test}}$.

**Defender's Assumptions.**    We closely mimic an assumption-free scenario similar to existing perturbation-based defenses. The scenario entails the knowledge-limited defender: (a) unaware whether a query is malicious or benign; (b) lacking prior knowledge of the strategy used by an attacker; and (c) perturbing each prediction independently (hence circumventing Sybil attacks). For added rigor, we also study attacker's countermeasures to our defense in Section 5.

## 4 APPROACH: MAXIMIZING ANGULAR DEVIATION BETWEEN GRADIENTS

**Motivation: Targeting First-order Approximations.**    We identify that the attacker eventually optimizes parameters of a stolen model $F(\cdot; \boldsymbol{w})$ (we drop the subscript $\cdot_A$ for readability) to minimize the loss on training examples $\{(\boldsymbol{x}_i, \tilde{\boldsymbol{y}}_i)\}$. Common to a majority of optimization algorithms is estimating the first-order approximation of the empirical loss, by computing the gradient of the loss

w.r.t. the model parameters $\boldsymbol{w} \in \mathbb{R}^D$:

$$\boldsymbol{u} = -\nabla_{\boldsymbol{w}} L(F(\boldsymbol{x}; \boldsymbol{w}), \boldsymbol{y}) \tag{1}$$

**Maximizing Angular Deviation (MAD).** The core idea of our approach is to perturb the posterior probabilities $\boldsymbol{y}$ which results in an adversarial gradient signal that maximally deviates (see Fig. 2) from the original gradient (Eq. 1). More formally, we add targeted noise to the posteriors which results in a gradient direction:

$$\boldsymbol{a} = -\nabla_{\boldsymbol{w}} L(F(\boldsymbol{x}; \boldsymbol{w}), \tilde{\boldsymbol{y}}) \tag{2}$$

to maximize the angular deviation between the original and the poisoned gradient signals:

$$\max_{\boldsymbol{a}} \ 2(1 - \cos\angle(\boldsymbol{a}, \boldsymbol{u})) = \max_{\hat{\boldsymbol{a}}} \ ||\hat{\boldsymbol{a}} - \hat{\boldsymbol{u}}||_2^2 \qquad (\hat{\boldsymbol{a}} = \boldsymbol{a}/||\boldsymbol{a}||_2, \hat{\boldsymbol{u}} = \boldsymbol{u}/||\boldsymbol{u}||_2) \tag{3}$$

Given that the attacker model is trained to match the posterior predictions, such as by minimizing the cross-entropy loss $L(\boldsymbol{y}, \tilde{\boldsymbol{y}}) = -\sum_k \tilde{y}_k \log y_k$ we rewrite Equation (2) as:

$$\boldsymbol{a} = -\nabla_{\boldsymbol{w}} L(F(\boldsymbol{x}; \boldsymbol{w}), \tilde{\boldsymbol{y}}) = \nabla_{\boldsymbol{w}} \sum_k \tilde{y}_k \log F(\boldsymbol{x}; \boldsymbol{w})_k = \sum_k \tilde{y}_k \nabla_{\boldsymbol{w}} \log F(\boldsymbol{x}; \boldsymbol{w})_k = \boldsymbol{G}^T \tilde{\boldsymbol{y}}$$

where $\boldsymbol{G} \in \mathbb{R}^{K \times D}$ represents the Jacobian over log-likelihood predictions $F(\boldsymbol{x}; \boldsymbol{w})$ over $K$ classes w.r.t. parameters $\boldsymbol{w} \in \mathbb{R}^D$. By similarly rewriting Equation (1), substituting them in Equation (3) and including the constraints, we arrive at our poisoning objective (Eq. 4-7) of our approach which we refer to as MAD. We can optionally enforce preserving accuracy of poisoned prediction via constraint (8), which will be discussed shortly.

$$\max_{\tilde{\boldsymbol{y}}} \ \left\| \frac{\boldsymbol{G}^T \tilde{\boldsymbol{y}}}{||\boldsymbol{G}^T \tilde{\boldsymbol{y}}||_2} - \frac{\boldsymbol{G}^T \boldsymbol{y}}{||\boldsymbol{G}^T \boldsymbol{y}||_2} \right\|_2^2 \qquad\qquad (= H(\tilde{\boldsymbol{y}})) \tag{4}$$

$$\text{where} \quad \boldsymbol{G} = \nabla_{\boldsymbol{w}} \log F(\boldsymbol{x}; \boldsymbol{w}) \qquad\qquad (\boldsymbol{G} \in \mathbb{R}^{K \times D}) \tag{5}$$

$$\text{s.t} \quad \tilde{\boldsymbol{y}} \in \Delta^K \qquad\qquad \text{(Simplex constraint)} \tag{6}$$

$$\text{dist}(\boldsymbol{y}, \tilde{\boldsymbol{y}}) \le \epsilon \qquad\qquad \text{(Utility constraint)} \tag{7}$$

$$\arg\max_k \tilde{\boldsymbol{y}}_k = \arg\max_k \boldsymbol{y}_k \qquad\qquad \text{(For variant MAD-argmax)} \tag{8}$$

The above presents a challenge of black-box optimization problem for the defense since the defender justifiably lacks access to the attacker model $F$ (Eq. 5). Apart from addressing this challenge in the next few paragraphs, we also discuss (a) solving a non-standard and non-convex constrained maximization objective; and (b) preserving accuracy of predictions via constraint (8).

**Estimating $\boldsymbol{G}$.** Since we lack access to adversary's model $F$, we estimate the jacobian $\boldsymbol{G} = \nabla_{\boldsymbol{w}} \log F_{\text{sur}}(\boldsymbol{x}; \boldsymbol{w})$ (Eq. 5) per input query $\boldsymbol{x}$ using a surrogate model $F_{\text{sur}}$. We empirically determined (details in Appendix E.1) choice of *architecture* of $F_{\text{sur}}$ robust to choices of adversary's architecture $F$. However, the *initialization* of $F_{\text{sur}}$ plays a crucial role, with best results on a fixed randomly initialized model. We conjecture this occurs due to surrogate models with a high loss provide better gradient signals to guide the defender.

**Heuristic Solver.** Gradient-based strategies to optimize objective (Eq. 4) often leads to poor local maxima. This is in part due to the objective increasing in all directions around point $\boldsymbol{y}$ (assuming $\boldsymbol{G}$ is full-rank), making optimization sensitive to initialization. Consequently, we resort to a heuristic to solve for $\tilde{\boldsymbol{y}}$. Our approach is motivated by Hoffman (1981), who show that the maximum of a convex function over a compact convex set occurs at the extreme points of the set. Hence, our two-step solver: (i) searches for a maximizer $\boldsymbol{y}^*$ for (4) by iterating over the $K$ extremes $\boldsymbol{y}_k$ (where $y_k{=}1$) of the probability simplex $\Delta^K$; and (ii) then computes a perturbed posterior $\tilde{\boldsymbol{y}}$ as a linear interpolation of the original posteriors $\boldsymbol{y}$ and the maximizer $\boldsymbol{y}^*$: $\tilde{\boldsymbol{y}} = (1 - \alpha)\boldsymbol{y} + \alpha\boldsymbol{y}^*$, where $\alpha$ is selected such that the utility constraint (Eq. 7) is satisfied. We further elaborate on the solver and present a pseudocode in Appendix C.

**Variant: MAD-argmax.** Within our defense formulation, we encode an additional constraint (Eq. 8) to preserve the accuracy of perturbed predictions. MAD-argmax variant helps us perform accuracy-preserving perturbations similar to prior work. But in contrast, the perturbations are *constrained* (Eq. 7) and are specifically introduced to maximize the MAD objective. We enforce the accuracy-preserving constraint in our solver by iterating over extremes of intersection of sets Eq.(6) and (8): $\Delta_k^K = \{\boldsymbol{y} \succeq 0, \mathbf{1}^T \boldsymbol{y} = 1, y_k \ge y_j, k \ne j\} \subseteq \Delta^K$.

## 5 EXPERIMENTAL RESULTS

### 5.1 EXPERIMENTAL SETUP

**Victim Models and Datasets.** We set up six victim models (see column '$F_V$' in Table 1), each model trained on a popular image classification dataset. All models are trained using SGD (LR = 0.1) with momentum (0.5) for 30 (LeNet) or 100 epochs (VGG16), with a LR decay of 0.1 performed every 50 epochs. We train and evaluate each victim model on their respective train and test sets.

**Attack Strategies.** We hope to broadly address all DNN model stealing strategies during our defense evaluation. To achieve this, we consider attacks that vary in query data distributions (independent and synthetic; see Section 3) and strategies (random and adaptive). Specifically, in our experiments we use the following attack models: (i) *Jacobian-based Data Augmentation* 'JBDA' (Papernot et al., 2017b);

| $F_V$ | Acc($F_V$) | Acc($F_A$) | | | |
|---|---|---|---|---|---|
| | | jbda | jbself | jbtop3 | k.off |
| MNIST (LeNet) | 99.4 | 89.2 | 89.4 | 87.3 | 99.1 |
| FashionMNIST (LeNet) | 92.0 | 38.7 | 45.8 | 68.7 | 69.2 |
| CIFAR10 (VGG16) | 92.0 | 28.6 | 20.7 | 73.8 | 78.7 |
| CIFAR100 (VGG16) | 72.2 | 5.3 | 2.9 | 39.2 | 51.9 |
| CUB200 (VGG16) | 80.4 | 6.8 | 3.9 | 21.5 | 65.1 |
| Caltech256 (VGG16) | 80.0 | 12.5 | 16.0 | 29.5 | 74.6 |

**Table 1: Victim models and Accuracies.** All accuracies are w.r.t undefended victim model.

(ii,iii) 'JB-self' and 'JB-top3' (Juuti et al., 2019); and (iv) *Knockoff Nets* 'knockoff' (Orekondy et al., 2019); We follow the default configurations of the attacks where possible. A recap and implementation details of the attack models are available in Appendix D.

In all attack strategies, the adversary trains a model $F_A$ to minimize the cross-entropy loss on a transfer set ($\mathcal{D}^{\text{transfer}} = \{(\boldsymbol{x}_i, \tilde{\boldsymbol{y}}_i)\}_{i=1}^{B}$) obtained by using the victim model $F_V$ to pseudo-label inputs $\boldsymbol{x}_i$ (sampled or adaptively synthesized). By default, we use $B$=50K queries, which achieves reasonable performance for all attacks and additionally makes defense evaluation tractable. The size of the resulting transfer set ($B$=50K examples) is comparable (e.g., 1× for CIFAR10/100, 2.1× for Caltech256) to size of victim's training set. In line with prior work (Papernot et al., 2016; Orekondy et al., 2019), we too find (Section 5.2.3) attack and defense performances are unaffected by choice of architectures, and hence use the victim architecture for the stolen model $F_A$. Due to the complex parameterization of VGG-16 (100M+), we initialize the weights from a pretrained TinyImageNet or ImageNet model (except for the last FC layer, which is trained from scratch). All stolen models are trained using SGD (LR=0.1) with momentum (0.5) for 30 epochs (LeNet) and 100 epochs (VGG16). We find choices of attacker's architecture and optimization does not undermine the defense (discussed in Section 5.2.3).

**Effectiveness of Attacks.** We evaluate accuracy of resulting stolen models from the attack strategies as-is on the victim's test set, thereby allowing for a fair head-to-head comparison with the victim model (additional details in Appendix A and D). The stolen model test accuracies, along with undefended victim model $F_V$ accuracies are reported in Table 1. We observe for all six victim models, using just 50K black-box queries, attacks are able to significantly extract victim's functionality e.g., >87% on MNIST. We find the knockoff attack to be the strongest, exhibiting reasonable performance even on complex victim models e.g., 74.6% (0.93×Acc($F_V$)) on Caltech256.

**How Good are Existing Defenses?** Most existing defenses in literature (Tramèr et al., 2016; Orekondy et al., 2019; Lee et al., 2018) perform some form of *information truncation* on the posterior probabilities e.g., rounding, returning top-$k$ labels; all strategies preserve the rank of the most confident label. We now evaluate model stealing attacks on the extreme end of information truncation, wherein the defender returns just the top-1 'argmax' label. This strategy illustrates a rough lower bound on the strength of the attacker when using existing defenses. Specific to knockoff, we observe the attacker is minimally impacted on simpler datasets (e.g., 0.2% accuracy drop on CIFAR10; see Fig. A5 in Appendix). While this has a larger impact on more complex datasets involving numerous classes (e.g., a maximum of 23.4% drop observed on CUB200), the strategy also introduces a significant perturbation ($L_1$=1±0.5) to the posteriors. The results suggest existing defenses, which largely the top-1 label, are largely ineffective at mitigating model stealing attacks.

**Defenses: Evaluation.** We evaluate all defenses on a non-replicability vs. utility curve at various operating points $\epsilon$ of the defense. We furthermore evaluate the defenses for a large query budget (50K). We use as *non-replicability* the accuracy of the stolen model on held-out test data $\mathcal{D}^{\text{test}}$.

**Figure 3: Attackers vs. Our Defense.** Curves are obtained by varying degree of perturbation $\epsilon$ (Eq. 7) in our defense. $\uparrow$ denotes higher numbers are better and $\downarrow$, lower numbers are better. Non-replicability objective is presented on the $x$-axis and utility on the $y$-axis.

We use two *utility* metrics: (a) accuracy: test-accuracy of the defended model producing perturbed predictions on $\mathcal{D}^{\text{test}}$; and (b) perturbation magnitude $\epsilon$: measured as $L_1$ distance $||\boldsymbol{y} - \tilde{\boldsymbol{y}}||_1$.

**Defense: Baselines.** We compare our approaches against three methods: (i) `reverse-sigmoid` (Lee et al., 2018): which softens the posterior distribution and introduces ambiguity among non-argmax probabilities. For this method, we evaluate non-replicability and utility metrics for the defense operating at various choices of their hyperparameter $\beta \in [0, 1]$, while keeping their dataset-specific hyperparameter $\gamma$ fixed (MNIST: 0.2, FashionMNIST: 0.4, CIFAR10: 0.1, rest: 0.2). (ii) `random noise`: For controlled random-noise, we add uniform random noise $\boldsymbol{\delta}_z$ on the logit prediction scores ($\tilde{\boldsymbol{z}} = \boldsymbol{z} + \boldsymbol{\delta}_z$, where $\boldsymbol{z} = \log(\frac{\boldsymbol{y}}{1-\boldsymbol{y}})$), enforce utility by projecting $\boldsymbol{\delta}_z$ to an $\epsilon_z$-ball (Duchi et al., 2008), and renormalize probabilities $\tilde{\boldsymbol{y}} = \frac{1}{1+e^{-\tilde{\boldsymbol{z}}}}$. (iii) `dp-sgd`: while our method and previous two baselines perturbs *predictions*, we also compare against introducing randomization to victim model *parameters* by training with the DP-SGD algorithm (Abadi et al., 2016). DP is a popular technique to protect the model against training data inference attacks. This baseline allows us to verify whether the same protection extends to model functionality.

## 5.2 RESULTS

In the follow sections, we demonstrate the effectiveness of our defense rigorously evaluated across a wide range of complex datasets, attack models, defense baselines, query, and utility budgets. For readability, we first evaluate the defense against attack models, proceed to comparing the defense against strong baselines and then provide an analysis of the defense.

### 5.2.1 MAD DEFENSE VS. ATTACKS

Figure 3 presents evaluation of our defenses MAD (Eq. 4-7) and MAD-argmax (Eq. 4-8) against the four attack models. To successfully mitigate attacks as a defender, we want the defense curves (colored solid lines with operating points denoted by thin crosses) to move *away from undefended* accuracies (denoted by circular discs, where $\epsilon$=0.0) *to ideal defense* performances (cyan cross, where Acc(Def.) is unchanged and Acc(Att.) is chance-level).

We observe from Figure 3 that by employing an identical defense across all datasets and attacks, the effectiveness of the attacker can be greatly reduced. Across all models, we find MAD provides reasonable operating points (above the diagonal), where defender achieves significantly higher test accuracies compared to the attacker. For instance, on MNIST, for <1% drop in defender's accuracy, our defense *simultaneously* reduces accuracy of the `jbtop3` attacker by 52% (87.3%→35.7%) and `knockoff` by 29% (99.1%→69.8%). We find similar promising results even on high-dimensional complex datasets e.g., on CUB200, a 23% (65.1%→41.9%) performance drop of `knockoff` for 2% drop in defender's test performance. Our results indicate effective defenses are achievable, where the defender can trade-off a marginal utility cost to drastically impede the attacker.

### 5.2.2 MAD DEFENSE VS. BASELINE DEFENSES

We now study how our approach compares to baseline defenses, by evaluating the defenses against the `knockoff` attack (which resulted in the strongest attack in our experiments). From Figure 4, we observe:

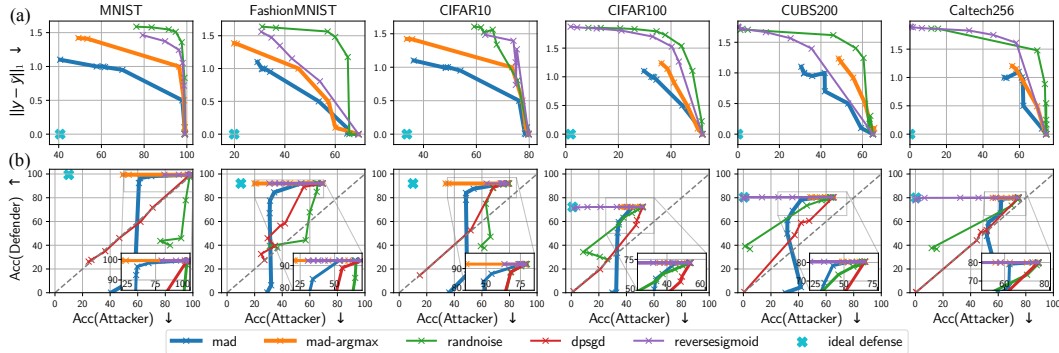

**Figure 4: Knockoff attack vs. Ours + Baseline Defenses** (best seen magnified). Non-replicability is presented on the $x$-axis. On $y$-axis, we present two utility measures: **(a) top:** Utility = $L_1$ distance **(b) bottom:** Utility = Defender's accuracy. Region above the diagonal indicates instances where defender outperforms the attacker.

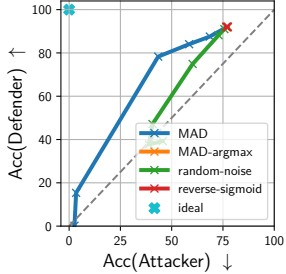

**Figure 5: Attacker argmax.** Follow-up to Figure 4b (CIFAR10), but with attacker using only the argmax label.

**Figure 6: Histogram of Angular Deviations**. Presented for MAD attack on CIFAR10 with various choices of $\epsilon$.

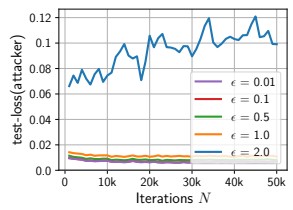

**Figure 7: Test loss.** Visualized during training. Colours and lines correspond to $\epsilon$ values in Fig. 6.

(i) *Utility objective = $L_1$ distance* (Fig. 4a): Although random-noise and reverse-sigmoid reduce attacker's accuracy, the strategies in most cases involves larger perturbations. In contrast, MAD and MAD-argmax provides similar non-replicability (i.e., Acc(Att.)) with significantly lesser perturbation, especially at lower magnitudes. For instance, on MNIST (first column), MAD ($L_1 = 0.95$) reduces the accuracy of the attacker to under 80% with $0.63\times$ the perturbation as that of reverse-sigmoid and random-noise ($L_1 \approx 1.5$).

(ii) *Utility objective = argmax-preserving* (Fig. 4b): By setting a hard constraint on retaining the label of the predictions, we find the accuracy-preserving defenses MAD-argmax and reverse-sigmoid successfully reduce the performance of the attacker by at least 20% across all datasets. In most cases, we find MAD-argmax in addition achieves this objective by introducing lesser distortion to the predictions compared to reverse-sigmoid. For instance, in Fig. 4a, we find MAD-argmax consistently reduce the attacker accuracy to the same amount at lesser $L_1$ distances. In reverse-sigmoid, we attribute the large $L_1$ perturbations to a shift in posteriors towards a uniform distribution e.g., mean entropy of perturbed predictions is $3.02 \pm 0.16$ (max-entropy = 3.32) at $L_1$=1.0 for MNIST; in contrast, MAD-argmax displays a mean entropy of $1.79 \pm 0.11$. However, common to accuracy-preserving strategies is a pitfall that the top-1 label is retained. In Figure 5 (see overlapping red and yellow cross-marks), we present the results of training the attacker using only the top-1 label. In line with previous discussions, we find that the attacker is able to significantly recover the original performance of the stolen model for accuracy-preserving defenses MAD-argmax and reverse-sigmoid.

(iii) *Non-replicability vs. utility trade-off* (Fig. 4b): We now compare our defense MAD (blue lines) with baselines (`rand-noise` and `dp-sgd`) which trade-off utility to mitigate model stealing. Our results indicate MAD offers a better defense (lower attacker accuracies for similar defender accuracies). For instance, to reduce the attacker's accuracy to <70%, while the defender's accuracy significantly degrades using `dp-sgd` (39%) and `rand-noise` (56.4%), MAD involves a marginal decrease of 1%.

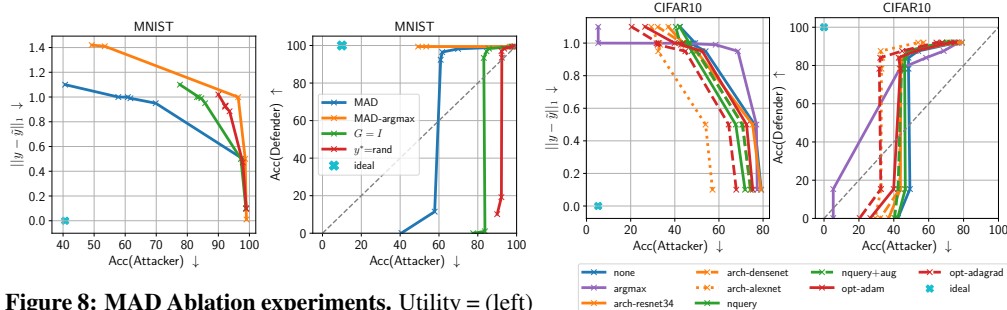

**Figure 8: MAD Ablation experiments.** Utility = (left) $L_1$ distance (right) defender test accuracy.

**Figure 9: Subverting the Defense.**

### 5.2.3 ANALYSIS

**How much angular deviation does MAD introduce?** To obtain insights on the angular deviation induced between the true and the perturbed gradient, we conduct an experiment by tracking the true gradient direction (which was unknown so far) at each training step. We simulate this by training an attacker model using online SGD (LR=0.001) over $N$ iterations using $B$ distinct images to query and a batch size of 1. At each step $t$ of training, the attacker queries a randomly sampled input $x_t$ to the defender model and backpropagates the loss resulting from $\tilde{y}_t$. In this particular experiment, the perturbation $\tilde{y}_t$ is crafted having exact knowledge of the attacker's parameters. We evaluate the angular deviation between gradients with ($a$) and without ($u$) the perturbation.

In Figure 6, we visualize a histogram of deviations: $\theta = \arccos \frac{u \cdot a}{\|u\|\|a\|}$, where $u = \nabla_w L(w_t, y, \cdot)$ and $a = \nabla_w L(w_t, \tilde{y}, \cdot)$. We observe: (i) although our perturbation space is severely restricted (a low-dimensional probability simplex), we can introduce surprisingly high deviations (0-115°) in the high-dimensional parameter space of the VGG16; (ii) for $\epsilon$ values at reasonable operating points which preserves the defender's accuracy within 10% of the undefended accuracy (e.g., $\epsilon \in [0.95, 0.99]$ for CIFAR10), we see deviations with mean 24.9° (yellow bars in Fig. 6). This indicates that the perturbed gradient on an average leads to a slower decrease in loss function; (iii) on the extreme end, with $\epsilon = \epsilon_{\max} = 2$, on an average, we find the perturbations successfully flips ($>90°$) the gradient direction leading to an increase on the test loss, as seen in Figure 7 (blue line). We also find the above observations reasonably transfers to a black-box attacker setting (see Appendix F.4), where the perturbations are crafted without knowledge of the attacker's parameters. Overall, we find our approach considerably corrupts the attacker's gradient direction.

**Ablative Analysis.** We present an ablation analysis of our approach in Figure 8. In this experiment, we compare our approach MAD and MAD-argmax to: (a) $G = I$: We substitute the jacobian $G$ (Eq. 5) with a $K \times K$ identity matrix; and (b) $y^*$=rand: Inner maximization term (Eq. 4) returns a random extreme of the simplex. Note that both (a) and (b) do not use the gradient information to perturb the posteriors.

From Figure 8, we observe: (i) poor performance of $y^*$=rand, indicating random untargeted perturbations of the posterior probability is a poor strategy; (ii) $G = I$, where the angular deviation is maximized between the posterior probability vectors is a slightly better strategy; (ii) MAD outperforms the above approaches. Consequently, we find using the gradient information (although a proxy to the attacker's gradient signal) within our formulation (Equation 4) is crucial to providing better model stealing defenses.

**Subverting the Defense.** We now explore various strategies an attacker can use to circumvent the defense. To this end, we evaluate the following strategies: (a) argmax: attacker uses only the most-confident label during training; (b) arch-*: attacker trains other choices of architectures; (c) nquery: attacker queries each image multiple times; (d) nquery+aug: same as (c), but with random cropping and horizontal flipping; and (e) opt-*: attacker uses an adaptive LR optimizer e.g., ADAM (Kingma & Ba, 2014).

We present results over the subversion strategies in Figure 9. We find our defense robust to above strategies. Our results indicate that the best strategy for the attacker to circumvent our defense

is to discard the probabilities and rely only on the most confident label to train the stolen model. In accuracy-preserving defenses (see Fig. 5), this previously resulted in an adversary entirely circumventing the defense (recovering up to $1.0\times$ original performance). In contrast, we find MAD is nonetheless effective in spite of the strategy, maintaining a 9% absolute accuracy reduction in attacker's stolen performance.

## 6 CONCLUSION

In this work, we were motivated by limited success of existing defenses against DNN model stealing attacks. While prior work is largely based on passive defenses focusing on information truncation, we proposed the first active defense strategy that attacks the adversary's training objective. We found our approach effective in defending a variety of victim models and against various attack strategies. In particular, we find our attack can reduce the accuracy of the adversary by up to 65%, without significantly affecting defender's accuracy.

**Acknowledgement.** This research was partially supported by the German Research Foundation (DFG CRC 1223). We thank Paul Swoboda and David Stutz for helpful discussions.

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

# Appendix

## A    OVERVIEW AND NOTATION

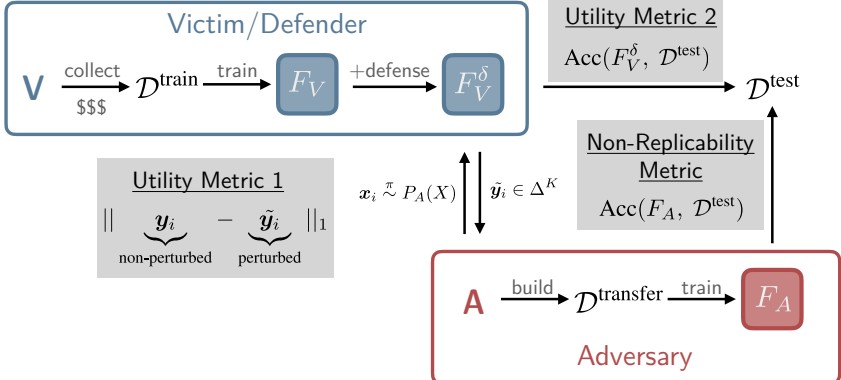

**Figure A1: Overview of Attack, Defense, and Evaluation Metrics.** We consider an attacker $A$ who exploits black-box access to defended model $F_V^\delta$ to train a stolen model $F_A$. In this paper, we take the role of the defender who intends to minimize replicability (i.e., $\text{Acc}(F_A, \mathcal{D}^{\text{test}})$), while maintaining utility of the predictions. We consider two notions of utility: (1) minimizing perturbations in predictions, measured here using $L_1$ distance; and (2) maintaining accuracy of the defended model on test set $\text{Acc}(F_V^\delta, \mathcal{D}^{\text{test}})$. Note that for a fair head-to-head comparison, we use the same held-out test set $\mathcal{D}^{\text{test}}$ to evaluate accuracies of both the defended model $F_V^\delta$ and stolen model $F_A$. Similar to all prior work, we assume $\mathcal{D}^{\text{train}}$, $\mathcal{D}^{\text{test}}$ are drawn i.i.d from the same (victim) distribution $\mathcal{D}_V$. Notation used in the above figure is further elaborated in Table A1.

|  |  |  |
|---|---|---|
|  | $\boldsymbol{x}$ | Inputs (images $\in \mathbb{R}^{C \times H \times W}$) |
|  | $\boldsymbol{y}, \tilde{\boldsymbol{y}}$ | Original, perturbed posterior predictions |
|  | $\Delta^K$ | Probability simplex over $K$ vertices |
| Attacker $A$ | $P_A(X)$ | Attacker's input data distribution |
|  | $\mathcal{D}^{\text{transfer}}$ | Transfer set ($= \{(\boldsymbol{x}_i, \boldsymbol{y}_i)\}$, where $\boldsymbol{x}_i \sim P_A(X)$, $\boldsymbol{y}_i = F_V(\boldsymbol{x}_i)$) |
|  | $F_A$ | Attacker's (stolen) model trained on $\mathcal{D}^{\text{transfer}}$ |
| Victim/Defender $V$ | $P_V(X)$ | Victim's input data distribution |
|  | $\mathcal{D}^{\text{train}}$ | Training data ($= \{(\boldsymbol{x}_i, \boldsymbol{y}_i)\}$, where $\boldsymbol{x}_i \sim P_V(X)$) |
|  | $F_V$ | Undefended model trained on $\mathcal{D}^{\text{train}}$ |
|  | $F_V^\delta$ | Defended model |
|  | $\mathcal{D}^{\text{test}}$ | Test set ($= \{(\boldsymbol{x}_i, \boldsymbol{y}_i)\}$, where $\boldsymbol{x}_i \sim P_V(X)$) |

**Table A1:** Notation

## B    RELATED WORK: EXTENSION

A summary of existing model stealing attacks and defenses is presented in Table A2.

## C    DETAILED ALGORITHM

We present a detailed algorithm (see Algorithm 1) for our approach described in Section 4.

The algorithm roughly follows four steps:

(i) **Predict (L2)**: Obtains posterior probability predictions $\boldsymbol{y}$ for input $\boldsymbol{x}$ using a victim model $F_V(\boldsymbol{x}; \boldsymbol{w}_V)$.

| | Black-box type | Proposed Attack | | Proposed Defense | | | |
|---|---|---|---|---|---|---|---|
| | | Input Query Data | Adapt.? | Strategy | P/D? | AP? | AC |
| 1. Lowd & Meek (2005) | Linear | Random Noise | ✓ | - | - | - | - |
| 2. Nelson et al. (2009) | Linear | Labeled Data | ✗ | Rejection | D | ✗ | 1 |
| 3. Nelson et al. (2010) | Linear | Random Noise | ✓ | - | - | - | - |
| 4. Alabdulmohsin et al. (2014) | Linear | Random Noise | ✓ | Ensembling | P | ✗ | 4 |
| 5. Tramèr et al. (2016) | Linear, NN | Random Noise | † | Rounding | P | ✓ | 5 |
| 6. Milli et al. (2018) | Linear, NN | Random Noise | ✓ | - | - | - | - |
| 7. Kesarwani et al. (2018) | Decision Tree | - | - | Detection | D | ✓ | 5 |
| 8. Chandrasekaran et al. (2019) | Linear | Random Noise | ✓ | Random Pert. | P | ✗ | - |
| 9. Papernot et al. (2017b) | CNN | Synth. Data | ✓ | - | - | - | - |
| 10. Correia-Silva et al. (2018) | CNN | Unlabeled Data | ✗ | - | - | - | - |
| 11. Pal et al. (2019) | CNN | Unlabeled Data | † | - | - | - | - |
| 12. Orekondy et al. (2019) | CNN* | Unlabeled Data | † | Rounding, Top-k | P | ✓ | 12 |
| 13. Jagielski et al. (2019) | CNN* | Unlabeled Data | ✓ | - | - | - | - |
| 14. Juuti et al. (2019) | CNN | Synth. Data | ✓ | Detection | D | ✓ | 9,14 |
| 15. Lee et al. (2018) | CNN | - | - | Reverse sigmoid | P | ✓ | 9 |
| 16. **Ours** | CNN* | - | - | Targeted Pert. | P | † | 9,12,14 |

**Table A2: Existing DNN Attacks and Defenses**. Complements the discussion in Section 2. 'CNN*': Complex ImageNet-like CNN. '†': Both. 'P/D': Perturbation/Detection. 'AP': Accuracy preserving (i.e., maintains top-1 labels of predictions). 'AC': Attacks considered.

(ii) **Estimate Jacobian $G$ (L3)**: We estimate a $\mathbb{R}^{K \times D}$ jacobian matrix on a surrogate model $F$. By default, we use as $F$ a randomly initialized model (more details in Appendix E.1). Each row of $G$ represents the gradient direction (in parameter space $\mathbb{R}^D$) over log likelihood of class $k$.

(iii) **Maximize MAD Objective (L4)**: We find the optimal direction $y^*$ which maximizes the MAD objective (Eq. 3). To compute the $\arg\max$, we iterative over the $K$ extremes of the probability simplex $\Delta^K$ to find $y^*$ which maximizes the objective. The extreme $y_k$ denotes a probability vector with $y_k = 1$.

(iv) **Enforce Utility Constraint (L5-7)**: We enforce the perturbation utility constraint (Eq. 7) by considering a linear interpolation of $y^*$ and $y$. The resulting interpolation probability vector $\tilde{y} := h(\alpha^*)$ represents the utility-constrained perturbed prediction that is returned instead of $y$.

---

1 **Function** PerturbedPredict-MAD($x$):
 **Input:** Input data $x$, model to defend $F_V(\cdot;\ w_V)$, proxy attacker model $F(\cdot;\ w)$
 **Output:** Perturbed posterior probability $\tilde{y} \in \Delta^K$ s.t. dist$(\tilde{y}, y) \leq \epsilon$
2  $\quad y := F_V(x;\ w_V)$                 // Obtain $K$-dim posteriors
3  $\quad G := \nabla_w \log F(x;\ w)$        // Pre-compute (K x D) Jacobian
4  $\quad y^* := \arg\max_{y_k \in \text{ext}(\Delta^K)} \left\| \frac{G^T y_k}{||G^T y_k||_2} - \frac{G^T y}{||G^T y||_2} \right\|_2^2$   // Alternatively ext($\Delta_k^K$)
   $\quad$ for MAD-argmax
5  $\quad$ Define $h(\alpha) = (1-\alpha)y + \alpha y^*$
6  $\quad \alpha^* := \arg\max_{\alpha \in [0,1], \text{dist}(\cdot) \leq \epsilon} \text{dist}(h(\alpha),\ y^*)$   // Find optimal step-size via
   $\quad$ bisection, or OptStep(.)  for $L_p$ norms
7  $\quad \tilde{y} := h(\alpha^*)$                     // Perturbed probabilities
8 **return** $\tilde{y}$
9
10 **Function** OptStep($y, y^*, \epsilon, p$):
11  $\quad \alpha^* := \max\left\{ \frac{\epsilon}{||y-y^*||_p},\ 1 \right\}$
12 **return** $\alpha^*$

**Algorithm 1: MAD Defense.** To supplement approach in Section 4

## D   ATTACK MODELS: RECAP AND IMPLEMENTATION DETAILS

**Jacobian Based Data Augmentation (`jbda`) (Papernot et al., 2017b).**   The motivation of the approach is to obtain a surrogate of the victim black-box classifier, with an end-goal of performing evasion attacks (Biggio et al., 2013; Goodfellow et al., 2014). We restrict discussions primarily to the first part of constructing the surrogate. To obtain the surrogate (the stolen model), the authors depend on an unlabeled 'seed' set, typically from the same distribution as that used to train the victim model. As a result, the attacker assumes (mild) knowledge of the input data distribution and the class-label of the victim.

The key idea behind the approach is to query perturbations of inputs, to obtain a reasonable approximation of the decision boundary of the victim model. The attack strategy involves performing the following steps in a repeated manner: (i) images from the substitute set (initially the seed) $\mathcal{D}$ is labeled by querying the victim model $F_V$ as an oracle labeler; (ii) the surrogate model $F_A$ is trained on the substitute dataset; (iii) the substitute set is augmented using perturbations of existing images: $\mathcal{D}_{\rho+1} = \mathcal{D}_\rho \cup \{\boldsymbol{x} + \lambda_{\rho+1} \cdot \mathrm{sgn}(J_F[F_A(\boldsymbol{x})]) \ : \ \boldsymbol{x} \in \mathcal{D}_\rho\}$, where $J$ is the jacobian function.

We use a seed set of: 100 (MNIST and FashionMNIST), 500 (CIFAR10, CUB200, Caltech256) and 1000 (CIFAR100). We use the default set of hyperparameters of Papernot et al. (2017b) in other respects.

**Jacobian Based {self, top-k} (`jbself`, `jbtop3`) (Juuti et al., 2019) .**   The authors generalize the above approach, by extending the manner in which the synthetic samples are produced. In `jbself`, the jacobian is calculated w.r.t to $k$ nearest classes and in `jb-self`, w.r.t the maximum a posterior class predicted by $F_A$.

**Knockoff Nets (`knockoff`) (Orekondy et al., 2019) .**   Knockoff is a recent attack model, which demonstrated model stealing can be performed without access to seed samples. Rather, the queries to the black-box involve natural images (which can be unrelated to the training data of the victim model) sampled from a large independent data source e.g., ImageNet1K. Consequently, no knowledge of the input data distribution nor the class-label space of the victim model is required to perform model stealing. The paper proposes two strategies on how to sample images to query: random and adaptive. We use the random strategy in the paper, since adaptive resulted in marginal increases in an open-world setup (which we have).

As the independent data sources in our `knockoff` attacks, we use: EMNIST-Letters (when stealing MNIST victim model), EMNIST (FashionMNIST), CIFAR100 (CIFAR10), CIFAR10 (CIFAR100), ImageNet1k (CUB200, Caltech256). Overlap between query images and the training data of the victim models are purely co-incidental.

We use the code from the project's public github repository.

**Evaluating Attacks.**   The resulting replica model $F_A$ from all the above attack strategies are evaluated on a held-out test set. We remark that the replica model is evaluated as-is, without additional finetuning or modifications. Similar to prior work, we evaluate the accuracies of $F_A$ on the victim's held-out test set. Evaluating both stolen and the victim model on the same test set allows for fair head-to-head comparison.

## E   SUPPLEMENTARY ANALYSIS

In this section, we present additional analysis to supplement Section 5.2.3.

### E.1   ESTIMATING $G$

Central to our defense is estimating the jacobian matrix $\boldsymbol{G} = \nabla_{\boldsymbol{w}} \log F(\boldsymbol{x}; \boldsymbol{w})$ (Eq. 5), where $F(\cdot; \boldsymbol{w})$ is the attacker's model. However, a defender with black-box attacker knowledge (where $F$ is unknown) requires determining $\boldsymbol{G}$ by instead using a surrogate model $F_{\text{sur}}$. We determine choice of $F_{\text{sur}}$ empirically by studying two factors: (a) *architecture of $F_{sur}$*: choice of defender's surrogate architecture robust to varying attacker architectures (see Fig. A2); and (b) *initialization of $F_{sur}$*: initialization of the surrogate model parameters plays a crucial role in providing a better defense. We

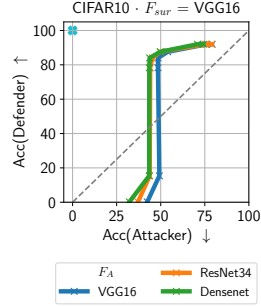

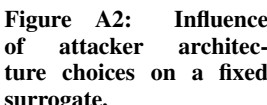

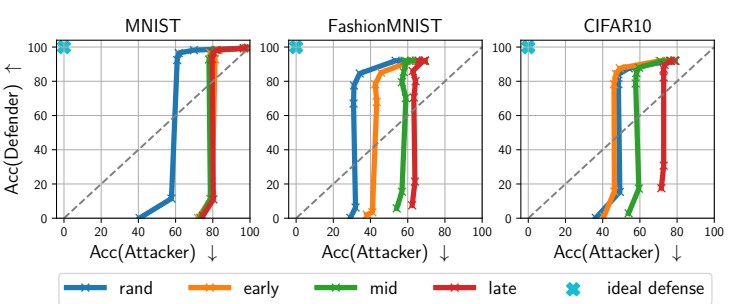

**Figure A2: Influence of attacker architecture choices on a fixed surrogate.**

**Figure A3: Influence of Initialization of a VGG16 Surrogate Model.** 'rand' = random initialization, ('early', 'mid', 'late') = $\sim(25, 50, 75)\%$ test accuracy of surrogate on test set.

|  | Undefended | MAD |
|---|---|---|
| MNIST | $0.88 \pm 14.41$ | $6.47 \pm 12.25$ |
| FashionMNIST | $0.89 \pm 15.76$ | $6.65 \pm 14.16$ |
| CIFAR10 | $1.93 \pm 13.02$ | $8.58 \pm 15.02$ |
| CIFAR100 | $2.15 \pm 18.82$ | $69.26 \pm 21.4$ |
| CUBS200 | $4.45 \pm 9.66$ | $446.93 \pm 23.87$ |
| Caltech256 | $4.93 \pm 21.25$ | $815.97 \pm 30.3$ |

**Table A3: Run times (in ms).** We report the mean and standard deviation of predictions of undefended and defended models, computed over 10K predictions.

consider four choices of initialization: {'rand', 'early', 'mid', 'late'} which exhibits approximately {chance-level 25%, 50%, 75%} test accuracies respectively. We observe (see Fig. A3) that a randomly initialized model, which is far from convergence, provides better gradient signals in crafting perturbations.

### E.2 RUN-TIME ANALYSIS

We present the run-times of our defended and undefended models in Table A3. The reported numbers were summarized over 10K unique predictions performed on an Nvidia Tesla V100. We find our optimization procedure Eq. (4-7) for all models take under a second, with at most 0.8s in the case of Caltech256. The primary computational bottleneck of our defense implementation is estimating matrix $\boldsymbol{G} \in \mathbb{R}^{K \times D}$ in Eq. 5, which currently requires performing $K$ (i.e., number of output classes) backward passes through the surrogate model. Consequently, we find that our inference times on Caltech256 can be further reduced to $0.3s \pm 0.04$ by using a more efficient surrogate architecture (e.g., ResNet-34).

## F ADDITIONAL PLOTS

### F.1 ATTACKER EVALUATION

We present evaluation of all attacks considered in the paper on an undefended model in Figure A4. Furthermore, specific to the `knockoff` attack, we analyze how training using only the top-1 label (instead of complete posterior information) affects the attacker in Figure A5.

### F.2 BUDGET VS. ACCURACY

We plot the budget (i.e., number of distinct black-box attack queries to the defender) vs. the test accuracy of the defender/attacker in Figure A6. The figure supplements Figure 1 and the discussion found in Section 5.2.1 of the main paper.

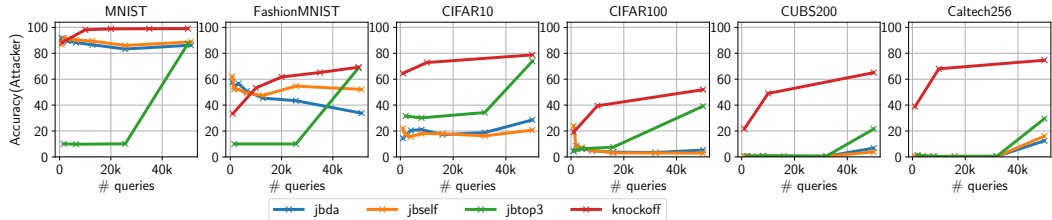

**Figure A4:** Evaluation of all attacks on undefended victim models.

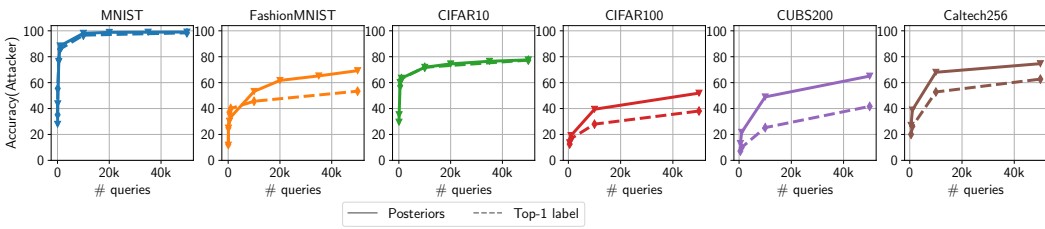

**Figure A5:** Stolen model trained using `knockoff` strategy on complete posterior information ($y$) and only the top-1 label of the posteriors ($\arg\max_k y_k$).

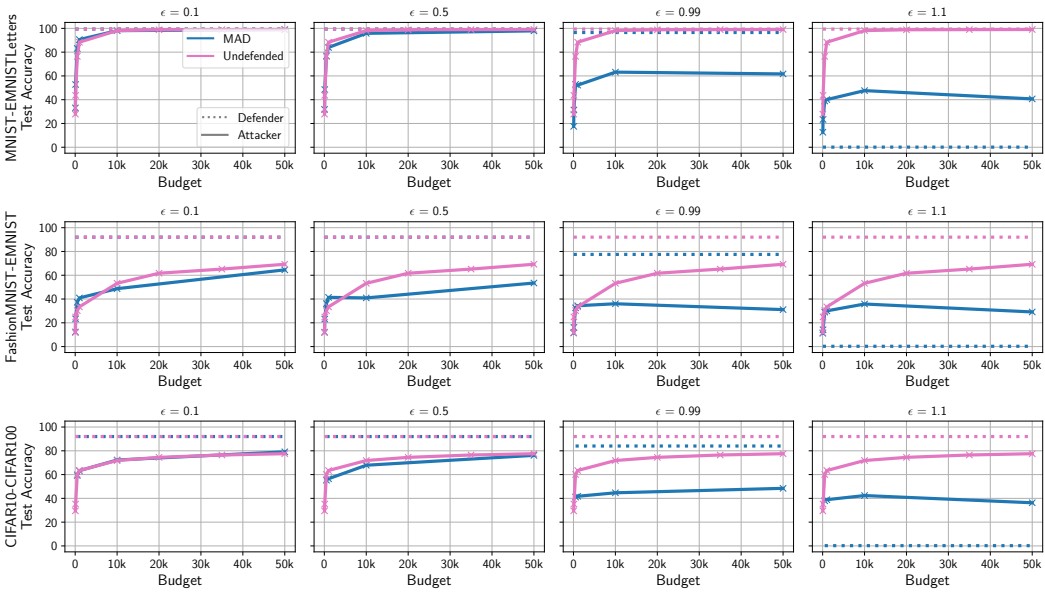

**Figure A6: Budget vs. Test Accuracy.** Supplements Fig. 3c in the main paper.

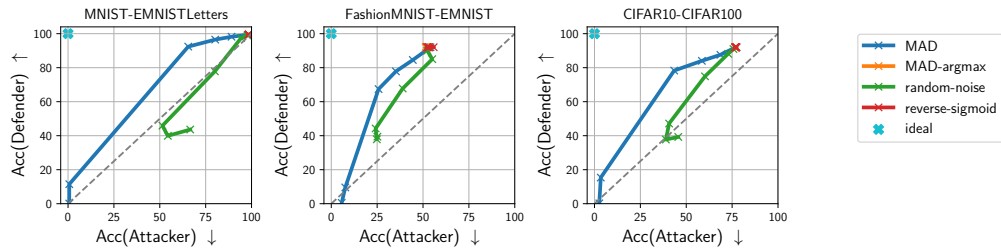

**Figure A7: Attacker argmax.** Supplements Fig. 4 in the main paper.

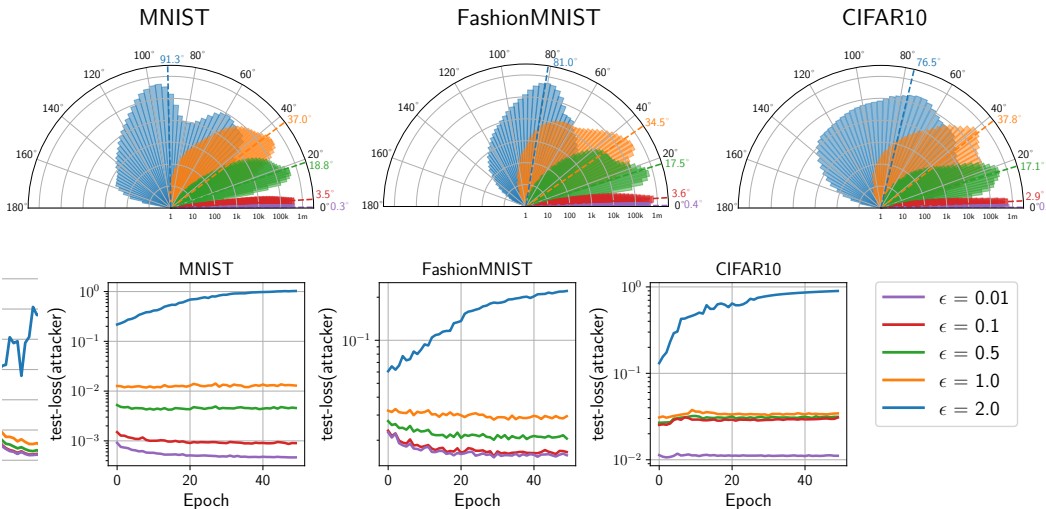

**Figure A8: Histogram of Angular Deviations (Black-box setting).** Supplements Fig. 6 in the main paper. The test-loss during of the attacker model for each of the histograms (over multiple $\epsilon$ values) are provided in the bottom row.

### F.3    ATTACKER ARGMAX

In Figure A7, we perform the non-replicability vs. utility evaluation (complementing Fig. 5 in the main paper) under a special situation: the attacker discards the probabilities and only uses the top-1 'argmax' label to train the stolen model. Relevant discussion can be found in Section 5.2.2.

### F.4    BLACK-BOX ANGULAR DEVIATIONS

In Figure A8, we provide the angular deviations obtained in a black-box setting over the course of training the attack model. We train the attacker model using the transfer set obtained by the knockoff approach (the strongest attacker in our experiments) for 50 epochs using a SGD (lr = 0.01, momentum = 0.5) and a batch size of 64. The experiment compliments our previous discussion in Section 5.2.3 of the main paper under "How much angular deviation does MAD introduce?". As before, we estimate the angular deviations as: $\theta = \arccos \frac{\boldsymbol{u} \cdot \boldsymbol{a}}{||\boldsymbol{u}||||\boldsymbol{a}||}$, where $\boldsymbol{u} = \nabla_{\boldsymbol{w}} L(\boldsymbol{w}_t, \boldsymbol{y}, \cdot)$ and $\boldsymbol{a} = \nabla_{\boldsymbol{w}} L(\boldsymbol{w}_t, \tilde{\boldsymbol{y}}, \cdot)$. We observe from Figure A8: (i) the defensive angular deviations introduced by MAD to posterior predictions transfer to a black-box attacker setting, when crafting perturbations without access to the adversary's model parameters; and (ii) although the setting introduces lower angular deviations at the extreme case of $\epsilon$=2.0 (e.g., $114.7° \rightarrow 76.5°$ in CIFAR10), we observe the perturbation sufficient to maximize the attacker's test loss. We find significant angular deviations introduced by our approach in a black-box setting as well.

### F.5    MAD ABLATION EXPERIMENTS

We present the ablation experiments covering all defender models in Figure A9. Relevant discussion is available in Section 5.2.3 of the main paper under "Ablative Analysis".

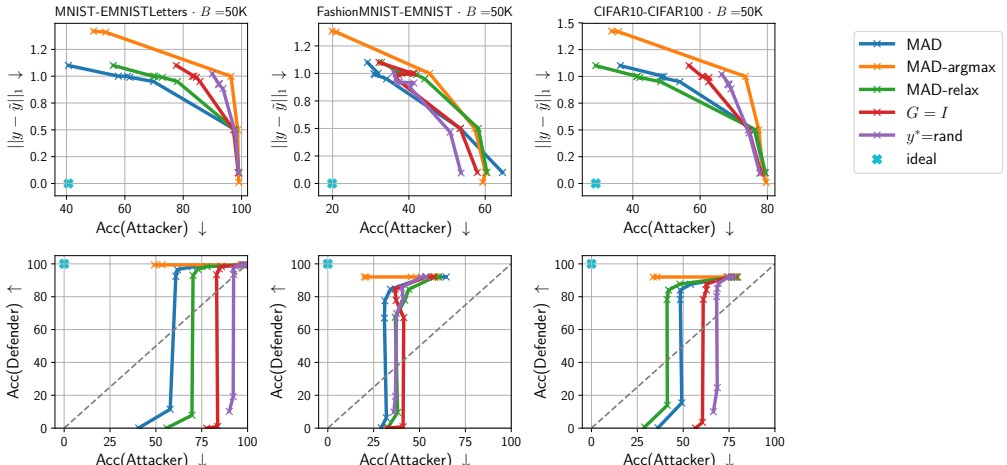

**Figure A9: MAD ablation experiments.** Supplements Fig. 8 in the main paper.

