# OpenReview forum: "Prediction Poisoning: Towards Defenses Against DNN Model Stealing Attacks"
_ICLR.cc/2020/Conference — Accept (Poster)_

### Official Review · AnonReviewer3 · 2019-10-23
**Official Blind Review #3**

**Rating:** 6

**Review:**

This paper aims at defending against model stealing attacks by perturbing the posterior prediction of a protected DNN with a balanced goal of maintaining accuracy and maximizing misleading gradient deviation. The maximizing angular deviation formulation makes sense and seemingly correct. The heuristic solver toward this objective is shown to be relatively effective in the experiments. While the theoretical novelty of the method is limited, the application in adversarial settings may be useful to advance of this research field, especially when it is relatively easy to apply by practitioners.I recommend toward acceptance of this paper even though can be convinced otherwise by better field experts.



**Experience Assessment:**

I do not know much about this area.

**Review Assessment: Checking Correctness Of Derivations And Theory:**

I assessed the sensibility of the derivations and theory.

**Review Assessment: Checking Correctness Of Experiments:**

I assessed the sensibility of the experiments.

**Review Assessment: Thoroughness In Paper Reading:**

I made a quick assessment of this paper.

---

> ### Author Response · Authors · 2019-11-12
> **Response to Review #3**
>
> We thank all reviewers for their valuable feedback. We are glad that reviewers find our “well-motivated” defense approach “effective” against model stealing, which “significantly supplement existing defenses”, and is further validated by “extensive experiments”. We are also pleased reviewers find the paper “well-written” and “very readable”.
>
> We appreciate AR3 for recognizing that our defenses “advances the research field”, especially due to a lack of an effective defense. While stronger theoretical results would certainly aid this line of study, our primary focus in this paper was establishing the first effective defense against a range of existing attack models and settings. We would be happy to answer any further questions.

---

### Official Review · AnonReviewer1 · 2019-10-23
**Official Blind Review #1**

**Rating:** 8

**Review:**

This paper proposed an effective defense against model stealing attacks.

Merits:
1) In general, this paper is well written and easy to follow.
2) The approach is a significant supplement to existing defense against model stealing attacks.
3) Extensive experiments.

However, I still have concerns about the current version.
I will possibly adjust my score based on the authors' response.

1) In the model stealing setting, attacker and defender are seemingly knowledge limited. This should be clarified better in Sec. 3.  It is important to highlight that the defender has no access to F_A, thus problem (4) is a black-box optimization problem for defense. Also, it is better to have a table to summarize the notations.

Additional questions on problem formulation:
a) Problem (4) only relies on the transfer set, where $x \sim P_A(x)$, right?
b) For evaluation metrics, utility and non-replicability, do they have the same D^{test}? How to determine them, in particularly for F_A?
c) One utility constraint is missing in problem (4). I noticed that it was mentioned in MAD-argmax, however, I suggest to add it to the formulation (4).

2)  The details of heuristic solver are unclear. Although the authors pointed out the pseudocode in the appendix, it lacks detailed analysis.

3) In Estimating G, how to select the surrogate model? Moreover, in the experiment, the authors mentioned that defense performances are unaffected by choice of architectures, and hence use the victim architecture for the stolen model. If possible, could the author provide results on different architecture choices for the stolen model as well as the surrogate model?

############## Post-feedback ################
I am satisfied with the authors' response. Thus, I would like to keep my positive comments on this paper. Although the paper is between 6 and 8, I finally decide to increase my score to 8 due to its novelty in formulation and extensive experiments.

**Experience Assessment:**

I have published in this field for several years.

**Review Assessment: Checking Correctness Of Derivations And Theory:**

I assessed the sensibility of the derivations and theory.

**Review Assessment: Checking Correctness Of Experiments:**

I assessed the sensibility of the experiments.

**Review Assessment: Thoroughness In Paper Reading:**

I read the paper at least twice and used my best judgement in assessing the paper.

---

> ### Author Response · Authors · 2019-11-12
> **Response to Review #1**
>
> We thank all reviewers for their valuable feedback. We are glad that reviewers find our “well-motivated” defense approach “effective” against model stealing, “significantly supplement existing defenses”, and further validated by “extensive experiments”. We are also pleased reviewers find the paper “well-written” and “very readable”.
>
> We now address individual concerns of AnonReviewer1.
>
> << (1a) Problem (4) relies on the transfer set, where $x \sim P_A(x)$, right?  >>
> - Yes. More precisely, problem (4) relies on a single input $x$ (queried by the adversary, sampled from an unknown $P_A(x)$).
>
>
> << (1b) Do utility and non-replicability have the same $D^{test}$? … How to determine $D^{test}$ for F_A?  >>
> - Yes. For evaluation purposes we use the same test set accuracies (of the corresponding dataset e.g., MNIST) to evaluate both the defended victim model ($F^{\delta}_V$) and attacker model ($F_A$). The setting allows for a fair head-to-head comparison of both models on a common test set.
> - [Revisions] We added an illustration (Fig. A1 of the appendix) and additional discussion (Appendix D “Evaluating attacks”) to better clarify this.
>
>
> << (1c) Utility constraint of MAD-argmax missing in (4) … suggest adding it to (4)  >>
> - Thanks for pointing this typo out. (4-7) previously presented our optimization problem for approach “MAD”.
> - [Revisions] Our revised version also presents the additional constraint for variant “MAD-argmax” in Eq. (8) .
>
>
> << (1d) Writing clarifications: (i) Better clarify that both attacker and defender are knowledge limited … (ii) highlight problem (4) is a black-box optimization problem for defense … (iii) Table to summarize notation >>
> - Thanks for the suggestions.
> - [Revisions] The revised manuscript addresses all the above:  (i) is clarified in Sec. 3 paragraphs “Knowledge-limited Attacker” and “Defender’s Assumptions” accordingly; (ii) is highlighted immediately after presenting problem (4); and (iii) We summarized the notation in Table A1 of the appendix.
>
>
> << (2) Details of the heuristic solver are unclear ...  Although the authors pointed out the pseudocode in the appendix, it lacks detailed analysis.  >>
> - We apologize for not making this clear.
> - [Revisions] We revised our paragraph “Heuristic Solver” in Sec. 4 of the manuscript and further elaborated on it in Appendix C.
>
>
> << (3) In estimating $G$, how to select the surrogate model? Results on different choices of architectures >>
> - We select the surrogate model based on empirical observations for choices of:
> (a) surrogate architectures: which has a negligible effect and is robust to choices of attacker architectures (Fig. A2); and
> (b) initialization of surrogate model: which plays a crucial role. Initializing the weights of the surrogate far from convergence (Fig. A3) provides a better gradient signal to poison the posteriors. Consequently, we choose a randomly-initialized model to estimate $G$.
> - [Revisions] We further clarify this in Section 4 (under “Estimating $G$”) of the revised manuscript and provide additional details (including results for choices of architectures and initializations) in Section E.1.

---

### Official Review · AnonReviewer4 · 2019-11-07
**Official Blind Review #4**

**Rating:** 3

**Review:**

The paper proposes a new method for defending against stealing attacks.

Positives:
1) The paper was very readable and clear.
2) The proposed method is straightforward and well motivated.
3) The authors included a good amount of experimental results.


Concerns:
1) You note that the random perturbation to the outputs performs poorly compared to your method, but this performance gap seems to decrease as the dataset becomes more difficult (i.e. CIFAR100). I’m concerned that this may indicate that the attackers are generally weak and this threat model may not be very serious. Overall, I’m skeptical of this threat model - the attackers require a very large number of queries, and don’t achieve great results on difficult datasets. Including results on a dataset like ImageNet would be nice.
2) How long does this optimization procedure take? It seems possibly unreasonable for the victim to implement this defense if it significantly lengthens the time to return outputs of queries.
3) Although this is a defense paper, it would be nice if the attacks were explained a bit more. Specifically, how are these attacks tested? You use the validation set, but does the attacker have knowledge about the class-label space of the victim? If the attacker trained with some synthetic data/other dataset, do you then freeze the feature extractor and train a linear layer to validate on the victim’s test set? It seems like this is discussed in the context of the victim in the “Attack Models” subsection, but it’s unclear what’s happening with the attacker.
4) It would be nice to see an angular histogram plot for a model where the perturbed labels were not crafted with knowledge of this model’s parameters - i.e. transfer the proposed defense to a blackbox attacker and produce this same plot. This would motivate the defense more.




**Experience Assessment:**

I have read many papers in this area.

**Review Assessment: Checking Correctness Of Derivations And Theory:**

I assessed the sensibility of the derivations and theory.

**Review Assessment: Checking Correctness Of Experiments:**

I assessed the sensibility of the experiments.

**Review Assessment: Thoroughness In Paper Reading:**

I read the paper at least twice and used my best judgement in assessing the paper.

---

> ### Author Response · Authors · 2019-11-12
> **(1/2) Response to Review #4**
>
> We thank all reviewers for their valuable feedback. We are glad that reviewers find our “well-motivated” defense approach “effective” against model stealing, “significantly supplement existing defenses”, and further validated by “extensive experiments”. We are also pleased reviewers find the paper “well-written” and “very readable”.
>
> We now address individual concerns of AnonReviewer4.
>
> << (1a) MAD is comparable to random-noise on CIFAR100 … Defense performance gap reduces as the dataset becomes more difficult >>
> - In terms of the two utility metrics we use to evaluate defense performance:
>   (i) *utility = defender’s accuracy* (y-axis of Fig. 4 - bottom; blue and green lines): the performance gap is nonetheless significant e.g., CIFAR100 MAD defender accuracy is 5% higher than random-noise at attacker accuracy (x-axis) = 44%; and
>   (ii) *utility = perturbation amount* (y-axis of Fig. 4 - top): we consistently find MAD significantly outperforms random-noise defense, even for difficult datasets e.g., CIFAR100 MAD defender introduces 1/3$\times$ lesser perturbation than random-noise at attacker accuracy = 44%.
> - [Revisions] We introduced magnification insets of dense regions in Fig. 4 of the revised manuscript to better convey the gaps.
>
>
> << (1b) Overall skeptical of the threat model: (i) requires large number of queries … (ii) attacker does not achieve great results >>
> - We believe the skepticism is unjustified, because:
>   (i) *Querying is cheap*: The bottleneck for stealing models is the *cost* of executing queries (e.g., money, latency) rather than the *number* of queries. The cost for querying in practise is cheap: 0.0015 USD per prediction on Google cloud prediction API for instance.
>   (ii) *Attack performance/results extracted per dollar is substantial*: Yes, the accuracy of attacker’s stolen models are imperfect (0.72-1.0$\times$ victim’s accuracy, Table 1). However, the dollars spent per accuracy point (“AP”) [1, 2, 3] for the attacker is a fraction of the victim’s. Consider Caltech256 for instance: (a) Victim = 11 USD per AP (22K images x 0.35 USD / 80 accuracy; 0.35 USD using Google’s data labeling platform) ignoring costs to collect and curate data, engineer the model, etc.; and (b) Attacker = 1 USD per AP (50K images x 0.0015 USD / 74.6 accuracy) by stealing.
> - Consequently, we argue that model stealing attacks pose a severe threat when viewing the problem as information extracted by the attacker per dollar.
>
>
> << (1c) Attack and defense results on ImageNet would be nice >>
> - While Imagenet evaluation would be interesting, the focus of our paper is defending existing attack models and on datasets that the attacks have proven to be effective. Unfortunately we are not aware of any existing model stealing attack on ImageNet, apart from a very recent arXiv paper [2] (Jagielski et al., Sep. 2019).
>
>
> << (2) How long does this optimization procedure take? … unreasonable if it significantly lengthens the time to return outputs of queries >>
> - We find all our optimization procedures take under a second. Specifically: 6ms (MNIST), 7ms (FashionMNIST), 9ms (CIFAR10), 69ms (CIFAR100), 0.4s (CUB200), 0.8s (Caltech256).
> - [Revisions] We added a discussion on run-times in Appendix E.2.
>
>
> << (3a) Would be nice if attacks were explained a bit more. Specifically, how are attacks tested? >>
> - Thanks for the suggestion. The attacks are evaluated on a common held-out victim test set. For a fair head-to-head comparison, the test set during evaluation is common to both the victim’s model and attacker’s stolen model.
> - [Revisions] Attacks are further clarified in the revised manuscript by (i) a visualization of attacker, defender and evaluation metrics in Appendix Figure A1; and (ii) extending our existing discussion on attack model details in Appendix D.
>
>
> << (3b) Does the attacker have knowledge about the class-label space of the victim? >>
> - All attackers are aware of the *number* of output classes of the victim model. As for the *semantics* of output class labels: (i) {knockoff}: does not require this knowledge; and (ii) {jbda, jb-self, jb-top3}: has the knowledge.
> - [Revisions] We clarified this in our existing discussion on attack models in Appendix E.
>
>
> << (3c) If the attacker is trained with some synthetic data/other dataset, do you then freeze the feature extractor and train a linear layer to validate on the victim’s test set? >>
> - No. We evaluate the attacker’s stolen model as-is on the victim’s test set. To further elaborate, we: (a) construct a transfer dataset (image-posterior pairs, where image = synthetic/other) using the attack strategies; (b) train the attack model $F_A$ using the transfer set; and (c) evaluate $F_A$ on the victim’s test set. Note that (a) and (b) are intertwined for some attacks.
> - [Revisions] We updated our manuscript to clarify this in Sec. 5 “Effectiveness of Attacks” and provided additional details in Appendix D “Evaluating attacks.''

---

> > ### Author Response · Authors · 2019-11-12
> > **(2/2) Response to Review #4**
> >
> > << (4) Angular histogram plot crafted without knowledge of model’s parameters .. would motivate the defense more >>
> > - We appreciate the suggestion. We find our defense similarly introduces angular deviations, even without knowledge of the attacker’s model parameters.
> > - [Revisions] We now indicate this in our discussion in the main manuscript. We provide further details in Appendix F.4.
> >
> > [1] Tramèr, Florian, et al. "Stealing machine learning models via prediction apis." USENIX 2016.
> > [2] Jagielski, Matthew, et al. "High-Fidelity Extraction of Neural Network Models." arXiv 2019.
> > [3] Anonymous authors. “Thieves on Sesame Street! Model Extraction of BERT-based APIs.” Under review at ICLR 2020.

---

> > ### Comment · AnonReviewer4 · 2019-11-13
> > **Reply to Reponse**
> >
> > Thanks for the responses. Here are points that still concern me:
> >
> > 1b.i)
> > Fair enough. I wasn't so concerned about the attacker not being able to perform this many queries (although you mention some previous defense methods do try to curb this), more that this illustrates that attackers are fairly weak as they require a lot of synthetic training data to get mediocre results.
> >
> > 1c)
> > I suspect attackers don't include ImageNet results because the results would be underwhelming. The attack methods seem very easy to adapt to new datasets, so it would have been nice to include this. I understand though that running ImageNet experiments in this short rebuttal period could be intractable.
> >
> > 2)
> > Thanks for including the time results. The Caltech256 are a little concerning though. The trend is that on more difficult datasets, the inference time scales very sharply with your defense (close to x200 for Caltech256). I imagine for practical datasets of interest, and a large number of queries, this could be a non-trivial bottleneck for anyone deploying this.
> >
> > 3a,b,c)
> > I appreciate the clarification on this.
> >
> > 4)
> > Thanks for including this. It provides credence to the method you employ.

---

> > > ### Author Response · Authors · 2019-11-15
> > > **Follow-up Response**
> > >
> > > We are happy our initial response helped address some concerns of AR4.
> > >
> > > Now, we hope to address the follow-up concerns.
> > >
> > >
> > > << 1.b.i (a) attackers are fairly weak .. get mediocre results >>
> > > - Stronger attackers would further motivate the model stealing threat. Model stealing is an active research topic (Table A2), and we find recent approaches demonstrate results to warrant a concern -- models require an immense amount of money and labour to develop, but can be substantially replicated (0.72-1.0$\times$ in our experiments) via prediction APIs for 10s-100s of dollars.
> > >
> > >
> > > << 1.b.i (b) attackers require a lot of synthetic data >>
> > > - The data is easy to obtain (e.g., random natural images) and comparable to victim’s training set size e.g., 1$\times$ (for CIFAR10/CIFAR100), 2.1$\times$ (Caltech256). Hence, we argue amount of data is not a limitation for attacks.
> > > - [Revisions] We remark on sizes of attacker’s transfer set relative to victim’s training set in Sec. 5 “Attack Strategies”.
> > >
> > >
> > > << 1c. Suspect Imagenet results would be underwhelming … attack methods seem very easy to adapt to new datasets … would be nice to include this >>
> > > - Extending methods to Imagenet would help clarify attack results. But, the experiments would indeed be intractable to perform in the given duration. We plan to tackle it in the future.
> > >
> > >
> > > << Caltech256 inference times a little concerning >>
> > > - We agree and we have already started working on this problem. Our current version reduces computation on Caltech256 to 0.3s by estimating $G$ using a more efficient surrogate architecture (ResNet-34 vs. VGG16 from before).
> > > - [Revisions] We indicate this in Appendix E.2. We will further discuss and update timing results.

---

### Author Response · Authors · 2020-04-26
**Code and Data**

The code and data is available here: https://resources.mpi-inf.mpg.de/d2/orekondy/predpoison/

---

### Decision · Program_Chairs · 2019-12-19

**Decision:**

Accept (Poster)

**Comment:**

The paper proposed an optimization-based defense against model stealing attacks.  A criticism of the paper is that the method is computationally expensive, and was not demonstrated on more complex problems like ImageNet.  While this criticism is valid, other reviewers seem less concerned by this because the SOTA in this area is currently focused on smaller problems.  After considering the rebuttal, there is enough reviewer support for this paper to be accepted.